# A Novel Method for the Evaluation of Bone Marrow Samples from Patients with Pediatric B-Cell Acute Lymphoblastic Leukemia—Multidimensional Flow Cytometry

**DOI:** 10.3390/cancers13205044

**Published:** 2021-10-09

**Authors:** Bettina Kárai, Katalin Tisza, Orsolya Eperjesi, Attila Csaba Nagy, Anikó Ujfalusi, Ágnes Kelemen, István Szegedi, Csongor Kiss, János Kappelmayer, Zsuzsanna Hevessy

**Affiliations:** 1Department of Laboratory Medicine, Faculty of Medicine, University of Debrecen, H-4032 Debrecen, Hungary; tisza.katalin@med.unideb.hu (K.T.); eperjesiorsi1348@gmail.com (O.E.); ujfalusi.aniko@med.unideb.hu (A.U.); kappelmayer@med.unideb.hu (J.K.); hevessy@med.unideb.hu (Z.H.); 2Faculty of Public Health, University of Debrecen, H-4002 Debrecen, Hungary; nagy.attila@sph.unideb.hu; 3Division of Pediatric Hematology/Oncology, Velkey László Child’s Health Center, H-3526 Miskolc, Hungary; kelemen.igyek@bazmkorhaz.hu; 4Department of Pediatrics, Faculty of Medicine, University of Debrecen, H-4032 Debrecen, Hungary; iszegedi@med.unideb.hu (I.S.); kisscs@med.unideb.hu (C.K.)

**Keywords:** pediatric acute lymphoid leukemia, flow cytometric examination, multidimensional dot-plots

## Abstract

**Simple Summary:**

By supporting the selection of the most suitable treatment protocol, the advancement of diagnostic methods contributes to achieving the best possible outcome for pediatric cases of acute lymphoblastic leukemia (ALL). In this study, we focused on a novel possibility in the flow cytometric (FC) analysis, as this method is the initial, crucial step in the diagnostic algorithm of ALL and can determine further diagnostic and therapeutic strategies. After the retrospective, multidimensional dot-plot-based FC analysis of 72 bone marrow samples of children with ALL, we found that the integrated appearance of immunophenotype resulted in a simple, quick, and accurate method. Furthermore, associations between immunophenotype and cytogenetic alterations were detected, which enabled the identification of cases with potential adverse outcome by completing the conventional FC analysis with multidimensional dot-plots. Standardized multi-center studies would be required to validate our results.

**Abstract:**

Multicolor flow cytometry (FC) evaluation has a key role in the diagnosis and prognostic stratification of ALL. Our aim was to create new analyzing protocols using multidimensional dot-plots. Seventy-two pediatric patients with ALL were included in this single-center study. Data of a normal BM sample and three BM samples of patients with BCP-ALL were merged, then all B cell populations of the four samples were presented in a single radar dot-plot, and those parameters and locations were selected in which the normal and pathological cell populations differed from each other the most. The integrated profile of immunophenotype resulted in a simple, rapid, and accurate method. There were no significant differences between the percentages of lymphoblasts in the detection of minimal residual disease (MRD) by multidimensional or conventional FC method (*p* = 0.903 at Day 15 and *p* = 0.155 at Day 33). Furthermore, we found associations between the position and the number of clusters of blast cells in the radar plots and cytogenetic properties (*p* = 0.002 and *p* < 0.0001 by the position and *p* = 0.02 by the number of subclones). FC analysis based on multidimensional dot-plots is not only a rapid, easy-to-use method, but can also provide additional information to screen cases which require detailed genetic examination.

## 1. Introduction

Acute lymphoblastic leukemia (ALL) is the most common pediatric malignant disease with an annual incidence of approximately 3.5/100,000 children worldwide; however, the overall 3-year survival rate can be as high as 87% if proper treatment is applied [1,2]. The 2016 modification of the World Health Organization (WHO) classification categories ALL based on recurrent genetic abnormalities which affect the characteristics of the disease and also determine the outcome and therapeutic decisions [3,4,5]. Besides genetic investigations, flow cytometry (FC) examination is a crucial part of establishing the diagnosis and later evaluating minimal residual disease (MRD) [6,7,8]. Furthermore, FC investigation, to some extent, is able to predict the presence of certain cytogenetic aberrations, as there are evidences of associations between the immunophenotype of pathological cells and cytogenetic aberrations, e.g., the absence of CD10 marker on abnormal lymphoblasts is associated with KMT2A gene rearrangement, and CD66c and CD25 positivity is associated with the presence of Philadelphia chromosome in ALL cases [9,10]. The expression of intracellular A subunit of coagulation factor XIII (FXIII-A) also needs to be mentioned here; though it is present in pathological B-cell lymphoblasts, the absence of this marker refers to the presence of a genetic subgroup belonging to the new provisional category of “B-other” genetic entity introduced by the WHO in 2016. Briefly, “B-other” group includes patients without recurrent genetic abnormalities, such as t(9;22)/BCR-ABL1, KMT2A (MLL) rearrangements, t(12;21)/ETV6-RUNX1, t(1;19)/TCF3-PBX1, or high hyperdiploidy [3,4,11,12]. The evaluation of MRD during induction therapy is proven to be an outstanding predictor of outcome, therefore it has a key role in risk-group stratification [13,14,15,16]. Even though the assessment of MRD was based on morphological examination initially [17], nowadays FC is recommended for this purpose, as—as proven by many working groups—it is as sensitive as a genetic method, and at the same time it is faster and more accessible. In order to become part of the diagnostic and follow-up protocols, FC method had to undergo thorough standardization [7,18,19,20,21]. The standardized FC method is based on well-defined antibody clones labelled with the recommended fluorochromes in 8 or 10-color settings, followed by conventional analysis, which uses bivariate dot-plots. In this case, we gain information about the expression of CD markers one by one, whether each antigen is expressed or not, and even if its presence is weak or bright positive [6]. The expression pattern of normal cells is well-known [22], so in each case, the interpreter compares the pattern observed to a normal one. This procedure requires remarkable experience, where the interpreters can recognize expression patterns and leukemia-associated immunophenotypes (LAIPs). In case of MRD detection, it is also a challenge to identify the abnormal cells in the regenerating bone marrow (BM), where all the stages of normal maturation are present during therapy; moreover, an immunophenotype shift of abnormal cells can also occur [23,24,25].

Due to the increasing number of fluorochromes and lasers in FC analyzers, more and more information can be obtained about the cells, and this resulted in the evolution of analyzing software. This way, besides the conventional analysis, a novel integrated, multidimensional method is also available, where all the applied antibodies of one tube can be examined at the same time in a single dot-plot. If the position of normal cell populations is known in the radar dot-plot, then the pathological cells can be identified due to their position difference compared to the normal cells. However, until now, only a few studies proved the usability of radar dot-plots for the evaluation of various hematological malignancies [26,27,28]. However, these studies either described maturing cells found in normal bone marrow [26] or used the multidimensional analysis to characterize a rare type of acute myeloid leukemia with a specific immunophenotype, the acute promyelocytic leukemia [27,28]. To the best of our knowledge, our work is the first to apply this multidimensional radar-plot analyzing method to characterize pediatric ALL samples.

Our aim was to exploit the benefits offered by multidimensional radar dot-plots in cases of pediatric ALL. We investigated whether they represent additional help in the diagnostic and follow-up workflow and if the multidimensional analysis with predefined gates could entirely eliminate the difficulties of the interpretation of bivariate dot-plots. This way, even flow experts with less experience could identify residual abnormal blast cells adequately. In addition, we also wanted to create an analyzing protocol which can identify lymphoblast populations with recurrent genetic abnormality, and we also inspected if there is a connection between the homogeneity of blast populations on the radar dot-plot and genetic alteration.

## 2. Materials and Methods

### 2.1. Study Design

The results of seventy-two children diagnosed with B-cell precursor ALL (BCP-ALL) were retrospectively processed. Multidimensional, radar dot-plots were optimized using flow cytometric analysis data from one normal and three randomly selected patients with BCP-ALL. Diagnostic BM samples of the patients were referred to our laboratory for multicolor FC examination between March 2013 and October 2020. The normal sample was a bone marrow follow up sample from a child in remission for more than 2 years. Sixty-nine children had BM samples of adequate quality for FC examination on Day 15, and sixty-three children on Day 33. Insufficient quality BM samples in which less than 30,000 syto positive events could be acquired and BM samples with low erythroid precursors (<2%) were excluded. The clinical and laboratory data are listed in Table 1.

The study was approved by the Scientific Research Ethical Committee of the Medical Research Council of Hungary (No IV/2103-1/2021/EKU) and was performed according to the 2008 Declaration of Helsinki. 

### 2.2. Immunophenotype Analysis

Bone marrow samples were routinely examined by eight-color labelling method. During the examined period, the combination of antibodies was slightly changed in accordance with the new guideline (Appendix A) [29]. Conventional FC analysis was performed by FACS Diva software v 8.0.3 (Becton Dickinson, San Jose, CA, USA). The raw data from the bi-dimensional FC examination was reanalyzed with novel protocols based on multidimensional dot-plots of Kaluza v 2.1 (Beckman Coulter, Brea, CA, USA) and Infinicyt v 2.0 (Cytognos, Salamanca, Spain) software.

### 2.3. Detection of Lymphoblasts in ALL by Multidimensional Dot-Plots

To examine the role of multidimensional dot-plots in the identification of pathological cells, BM of children with BCP-ALL were analyzed by a novel analyzing protocol using radar dot-plots of Kaluza software. The software allows selecting of the number and the position of examined parameters, which influenced the position of cell populations on radar dot-plots. To discriminate between normal and pathological cells, radar dot plots had to be optimized. First, data of a normal BM sample and three BM samples of patients with BCP-ALL were merged in order to have a data pool containing pathological B cells and hematogones of the normal B cell maturation. Normal and pathological B cells were gated on bivariate dot-plots. Then, all B cell populations of the four samples were presented in a single radar dot-plot, and those parameters and locations were selected in which the normal and pathological cell populations differed from each other the most. One radar dot-plot was optimized for each tube. To identify pathological cells with radar dot plot, the gating strategy was the following: living cells were selected on the basis of syto staining and light scatter character. Then, CD19 positive B cells were gated and presented on multidimensional dot-plots. Pathological cells were easily identified as normal and pathological cells differed unequivocally on the optimized radar dot-plots (Figure 1). Finally, lymphoblasts gated on radar dot-plots were displayed on bivariate dot-plots to examine the expression of certain markers individually. The detailed immunophenotype was necessary in order to test that the population with a position different from normal cells was really the pathological population. The abnormal lymphoblasts may resemble normal hematogones (CD10/CD34/nTdT positive), but there are several signs that their immunophenotype may be different from the normal: LAIPs might be detected. Examples include overexpression of CD10, absence of CD45, CD34, or CD38, presence of myeloid markers (CD66c, CD11a, and CD123) or abnormal maturation pattern in the CD10-CD20; CD19-CD45; and CD19-CD10 bivariate dot plots.

In case of de novo samples, the position of pathological cells was examined in more detail. The area of radar dot-plots which differed from the normal was divided into six parts by Tube 1 (Figure 2A) and four parts by Tube 2 (Figure 2B). BCP-ALL cases were categorized by the number of the part where the lymphoblasts were located. 

Then, we compared the position of lymphoblasts with cytogenetic alterations. Patients with *KMT2A* gene rearrangement, iAMP21, t(1;9), and t(1;19) were excluded from the study due to the low number of cases. 

Following the analysis of de novo BM samples, we focused on the detection of pathological lymphoblasts in MRD samples on Day 15 and Day 33. Each sample was analyzed by two persons, a junior resident, and an experienced laboratory specialist. The percentage of lymphoblasts identified by the novel multidimensional protocol were compared to the percentage of lymphoblasts determined by conventional flow cytometric examination. Previous results of FC-MRD obtained by conventional bivariate analysis were not known when samples were analyzed with the new multidimensional analysis system. The two independent investigators were also unaware of the results of each other’s analysis in the MRD study. 

In the second part of our study, de novo samples were analyzed by Infinicyt software. Following the identification of lymphoblasts by conventional examination, pathological cells were displayed on automatic population separator (APS) dot plots. APS dot plots utilize all the information about the light scatter character and the expression of antibodies placed in a tube to separate the clusters from each other [30]. The homogeneity of the pathological lymphoblasts was examined using these APS dot-plots (Figure 3). Then, the association between the number of clusters and the genetic character of the pathological population was investigated. 

### 2.4. Statistical Analysis

The normal distribution was tested by the Shapiro–Wilk test. To compare two paired groups, we used paired Student’s t-test for parametric and Wilcoxon test for non-parametric data. To check the association between two variables, Spearman´s correlation was applied. Dichotomous categorical variables were compared using Pearson´s Chi square test. Two Proportion Z-Test was used to compare proportions. Statistical analysis and the creation of figures were performed using SPSS 20.0 (Chicago, IL, USA), STATA 14.2 (College Station, TX, USA) and GraphPad Prism 6.0 (San Diego, CA, USA) statistical programs.

## 3. Results

### 3.1. Comparison of Pathological Cell Detection and Enumeration with Multidimensional and Bivariate Dot-Plots in ALL

Pathological cells were categorized similarly in MRD samples based on conventional and multidimensional dot-plots in terms of determining the percentage of pathological cells (*p* = 0.903 at Day 15 and *p* = 0.155 at Day 33). The correlations between the two analyzing methods were stronger on Day 15 compared to Day 33 (*p* < 0.0001, r = 0.944 at Day 15 and r = 0.571 at Day 33). 

In the Day 15 samples, pathological and normal cells can be distinguished well as there are no normal hematogones in these samples only mature B-cells and pathological lymphoblasts can be detected (Figure 4). Furthermore, we found that if one marker was changed in the panel (CD66c + CD123 for CD66c and CD73 + CD303 for CD123; Appendix A), it did not influence the discrimination between normal and pathological B cells. However, in the case of bivariate dot-plots of Day 33 samples, the gating of blast cells required more attention due to the regeneration of the BM. 

There was no significant difference between the two examiners in terms of determining the percentages of Day 15 and Day 33 pathological cells with multidimensional analysis by both tubes (*p* = 0.060 and 0.233 by Tubes 1 and 2 on Day 15; *p* = 0.189 and *p* = 0.283 by Tubes 1 and 2 on Day 33) (Figure 5). The correlation between the two examiners proved to be strong according to Spearman’s correlation; the *p* value was lower than 0.0001 by all comparisons (r = 0.975 and r = 0.953 by Tubes 1 and 2 on Day15; r = 0.650 and r = 0.798 by Tubes 1 and 2 on Day 33). 

### 3.2. Association between the Position and the Homogeneity of Pathological Cells and Cytogenetic Alterations

The position of the pathological cells showed association with the recurrent cytogenetic abnormalities (*p* = 0.002 by Tube 1 and *p* < 0.0001 by Tube 2). Three cytogenetic categories—patients with hyperdiploidy (*n* = 22), patients with t(12;21) translocation (*n* = 18), and patients categorized as the “*B-other*” subgroup (*n* = 18)—had a sufficiently high number of samples to examine the association in more detail. Regarding Tube 1, lymphoblasts were located in the first pre-defined gate more frequently by patients with hyperdiploidy compared to patients with t(12;21) translocation (*p* = 0.009). While lymphoblasts in cases with t(12;21) translocation usually appeared in the second pre-defined gate compared to cases with hyperdiploidy (*p* = 0.002) or the “*B-other*” subgroup (*p* = 0.001). As for Tube 2, the position of lymphoblasts were clearer. Lymphoblasts in hyperdiploidy cases appeared significantly more frequently in the third gate (*p* = 0.002 as compared to t(12;21) and *p* = 0.009 compared to the “*B-other*” subgroup). Lymphoblasts in t(12;21) translocation cases were often located in the first gate (*p* < 0.001 and *p* = 0.002, in comparison with cases with hyperdiploidy and “*B-other*” subgroup, respectively). Finally, pathological cells in cases categorized in the “*B-other*” subgroup usually appeared in the fourth gate (*p* = 0.001 and *p* = 0.048, in comparison with cases with t(12;21) translocation and hyperdiploidy, respectively) (Figure 6). 

By examining the detailed immunophenotype of the lymphoblasts at a given position, we found that no specific immunophenotype-pattern could be associated with the positions. CD66c is more often positive in positions 1 and 3. CD34 is mostly negative in positions 4 and 6, while it is more often positive in other positions. CD10 is brighter in positions 2, 3, and 4 than in positions 1, 5, and 6. The CD38 was positive in all cases in position 5, and decreased in intensity in the others. CD45 was more intensely expressed on lymphoblasts in positions 5 and 6. For tube two, CD58 was mostly positive in all positions. The highest expression was expressed on lymphoblasts at position 2. CD123 was more frequently expressed on lymphoblasts located in position 3 or 4. CD33 expression on lymphoblasts in position 2 did not occur in any of the cases. Finally, CD81 expression was more frequently observed in lymphoblasts located in position 4, with normal bright intensity (Table 2).

The tables represent the percentage of cases in which the expression of each marker is shown in the different blast positions using conventional bivariate analysis.

Furthermore, we found significant correlation between the cytogenetic properties and the number of clusters of the pathological cells (*p* = 0.02) (Figure 7). In the cases of the patients of the “*B-other*” subgroup, we realized that the pathological cells do not form a homogenous population, unlike in the cases of the patients with hyperdiploidy (*p* = 0.011) or t(12;21) translocation (*p* = 0.048). 

## 4. Discussion

Evaluation of MRD is a powerful prognostic factor in pediatric B-cell acute lymphoblastic leukemia and is being considered as a basis for deintensification or escalation in treatment protocols [13,14,15,16,17]. MRD can be assessed by flow cytometry and molecular genetics, with the former being a rapid, relatively inexpensive, and widely applicable method [16,20,21]. With the acquisition of several million cells in the next generation flow technique [31] the sensitivity of the method became comparable to the molecular methods [29,32]. Increasing the number of colors in a tube and the number of acquired events the sensitivity of FC can reach 2-in-10^6^ (0.0002%) [33]. On the other hand, analysis and interpretation of FC data requires a well-trained, skilled, and experienced interpreter who not only knows the normal immunophenotype patterns of B-cell maturation in rest and marrow regeneration and can differentiate pathological populations from regenerating hematogones, but also holds and applies a knowledge of hematological diseases, their classification and therapy, as the nature of therapy may affect antigen expression, a phenomenon especially important to take into consideration when a patient is under a targeted therapy [34].

The novel analyzing protocol described in our study applied two multidimensional dot-radar plots, which can provide information about all the examined markers at the same time. The integrated appearance of the immunophenotype simplifies the identification of pathological cells. In the case of conventional bivariate flow cytometric analysis, the examiner first has to learn the immunophenotype of normal hematogones and the feature of pathological cells in the different diseases. Then, the detected immunophenotype of pathological cells is compared to the learned patters. In case of multidimensional dot plots, it is not required to know all expression changes of antigens during maturation; it is enough to learn the position of normal cell populations. We found that the correlations between the two analyzing methods were stronger on Day 15 compared to Day 33 MRD samples, as in the case of bivariate dot-plots of Day 33 samples, the regeneration of the BM and the presence of normal hematogones makes it difficult to identify blasts. However, with the application of the novel multidimensional analysis protocol there was no significant difference between the beginner and the experienced examiner in terms of determining the percentages of pathological cells in Day 15 and Day 33 BM samples.

Pattern-guided, semi-automated or automated analyses could be a promising tool for standardized interpretation of de novo and MRD samples in hematological malignancies [35]. The development of software’s enabled the automation of certain steps of the detection of pathological cells by multidimensional dot-plots [36,37]. Through a mathematical calculation based on Principal Components Analysis (PCA), the immunophenotype determines the position of pathological cells in multidimensional dot-plots. With automated cluster analysis, not only are the determination of major cell populations possible, but small subsets also become identifiable (e.g., Infinicyt, FlowSom). Our results obtained with Infinicyt software confirmed that the population which appeared to be homogenous on bivariate dot-plots could be divided into several subclones by automated cluster separation, which could provide more information about a certain case. As the presence of heterogeneous lymphoblast populations occurred more frequently in patients assigned to the “*B-other*” subgroup compared to patients with recurrent cytogenetic alteration associated with good prognosis (hyperdiploidy, t(12;21)), multidimensional analysis may have prognostic significance, too. Furthermore, newly identified associations between immunophenotype and recurrent cytogenetic alteration could be observed, which supports the view that FC analysis might play a greater role in the diagnostic and prognostic examination of ALL. 

The most important finding of our study was that FC analysis based on multidimensional dot-plots was able to not only corroborate the results of the conventional FC analysis, but also provide additional information. 

The major limitation of multidimensional analysis is that standardized antibody combinations are required. Nevertheless, we found that if only one marker was changed, especially a LAIP one, it did not influence the integrated visualization-based analysis. As it was a single-center study, only a limited number of cases could be included, therefore the cases with rare cytogenetic alterations had to be excluded from the study.

## 5. Conclusions

In conclusion, the role and application of flow cytometric analysis in establishing the diagnosis and prognosis of ALL is changing. Multidimensional dot plots can simplify the analysis and support the determination of the accurate percentage of lymphoblasts even with a less experienced interpreter and in difficult cases where there are less LAIPs. Furthermore, complementing the conventional analysis with multidimensional dot plots can help to identify those cases which require examination in more detail with genetic methods. 

## Figures and Tables

**Figure 1 cancers-13-05044-f001:**
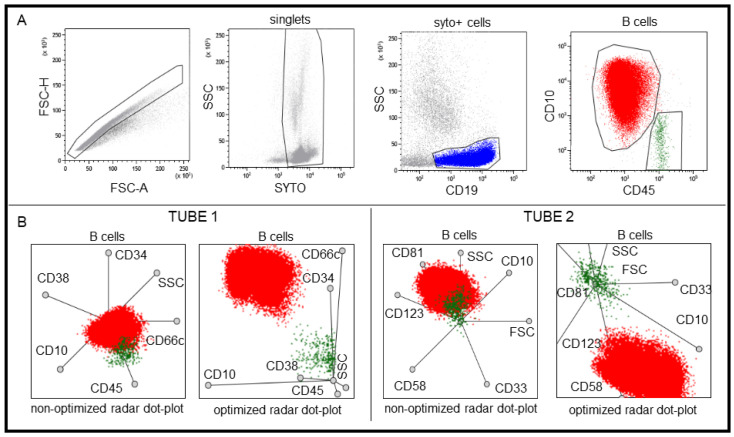
Steps to optimize the radar plot. In bivariate dot plots (Kaluza software) first singlets, then syto positive nucleated cells and finally CD19 positive B cells were gated. (**A**) In de novo samples, gates were applied around the normal, mature (green) CD45+/CD10- B cells and the pathological (red) CD45-/CD10++ B cells. (**B**) Normal and pathological B cells are not properly separated in multidimensional plots when the axes are of equal length and are evenly spaced in the non-optimized version, however, cell populations are well separated if the length and position of the axes are optimized.

**Figure 2 cancers-13-05044-f002:**
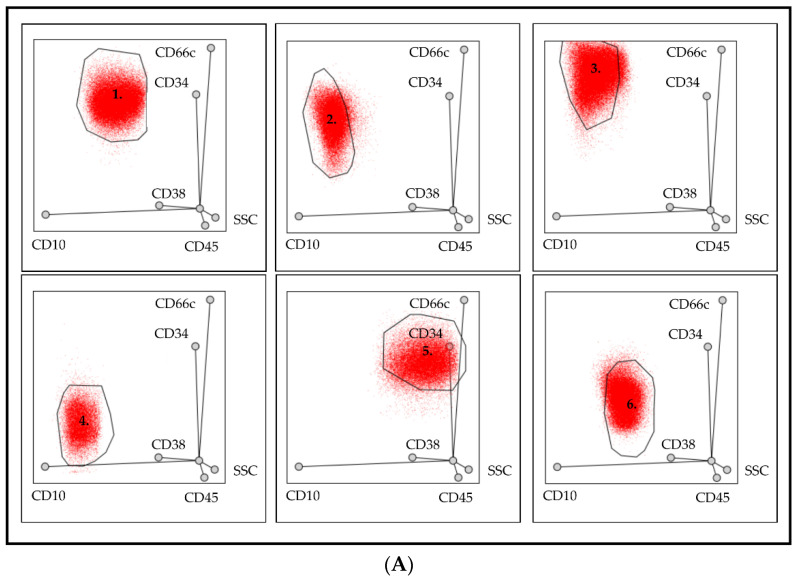
The most common positions of pathological lymphoblasts by Tube 1 (**A**) and Tube 2 (**B**). There were six (**A**) and four (**B**) predilection positions where the pathological lymphoblast were located. On the basis of the percentage of lymphoblasts in these predefined gates, all cases were assigned to one of the typical patterns.

**Figure 3 cancers-13-05044-f003:**
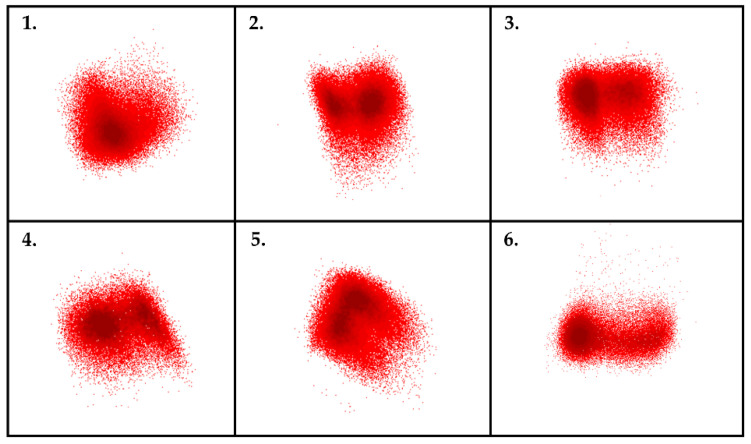
Examination of the homogeneity of pathological lymphoblasts. Representative density dot plots of a homogenous (**1**) and heterogeneous lymphoblast populations (**2**–**6**) on APS dot-plot. Color intensity is proportional to the number of cells. APS dot plot automatically tries to separate populations by looking at all markers together. For a heterogeneous lymphoblast population (**2**–**6**), it is clearly visible that there are two dark red areas, so a single population on a bivariate dot plot can be separated into further subpopulations.

**Figure 4 cancers-13-05044-f004:**
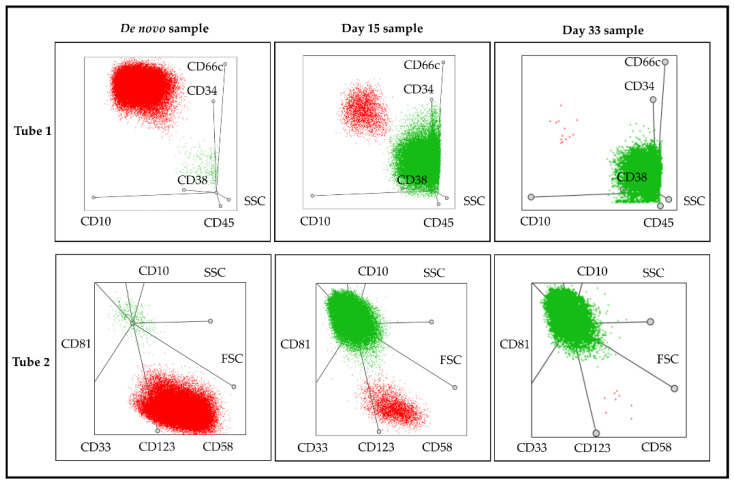
Representative multidimensional dot-plots of a patient with BCP-ALL. The pathological lymphoblasts (red color) and normal B cells (green color) separated from each other on the basis of their positions on multidimensional dot-plots. Note the decreasing trend of pathological cells and the increasing proportion of normal B cells as therapy progresses. Details of the antibodies in each tube can be found in Figure 1.

**Figure 5 cancers-13-05044-f005:**
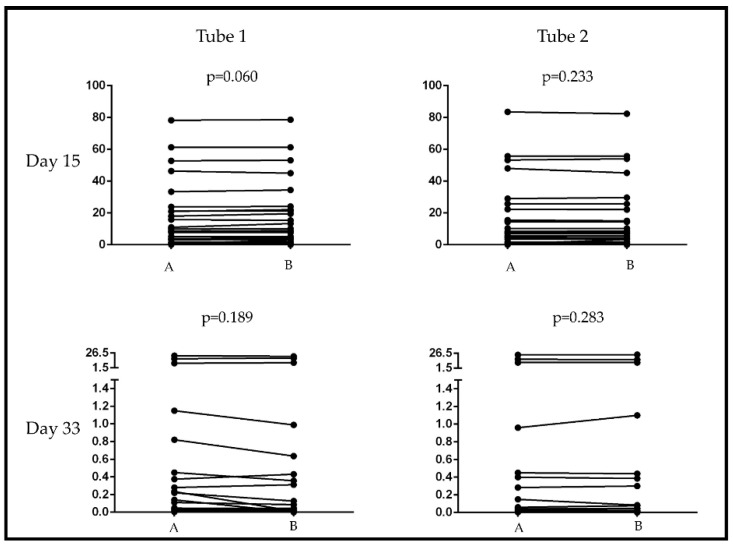
Comparison of percentages of Day 15 and Day 33 pathological cells determined by two examiners with multidimensional analysis. There were no significant differences between the beginner (A) and experienced (B) examiners. Details of the antibodies in each tube can be found in Appendix A.

**Figure 6 cancers-13-05044-f006:**
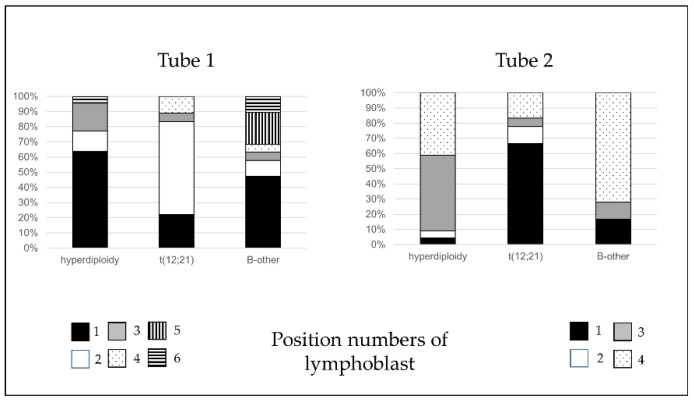
Association of the position of lymphoblasts on radar dot plots and recurrent genetic alterations. The position of lymphoblasts by both tubes was found to be significantly associated with recurrent genetic alterations. Details of the antibodies in each tube can be found in Appendix A.

**Figure 7 cancers-13-05044-f007:**
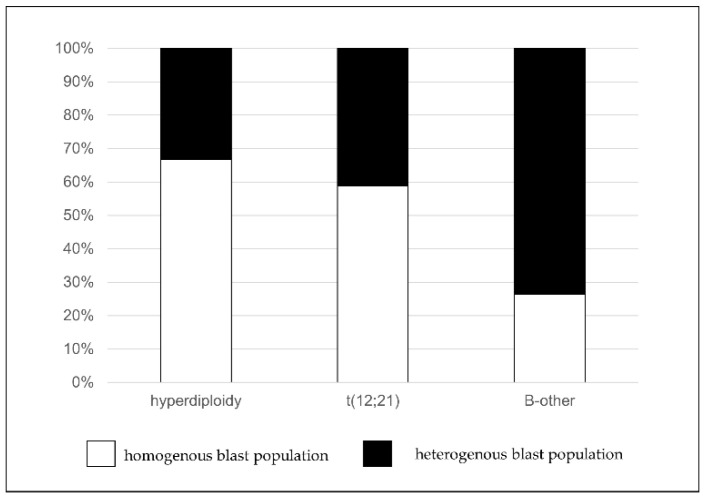
Association of the heterogeneity of lymphoblasts and recurrent genetic alteration. Lymphoblast of patients assigned to the “*B-other”* subgroup proved to be significantly more heterogeneous than those of patients with good prognostic genetic alterations (hyperdiploidy and t(12;21)).

**Table 1 cancers-13-05044-t001:** Clinical and laboratory data of children with ALL.

Variables	N	%
**Gender**		
Male	37	51.4
Female	35	48.6
**Age**		
<6 years	42	58.3
≥6 years	30	41.7
**Cytogenetic categories**		
high hyperdiploidy	22	30.5
t(12;21) (p13.2;q22.1)	18	25.0
t(1;19) (q23;p13.3)	3	4.2
“B-other” subgroup	19	26.4
*KMT2A* rearrangement	1	1.4
iAMP21	2	2.8
t(9;22) (q34.1;q11.2)	6	8.3
t(1;9) (q24;q34)	1	1.4
**FC-MRD (Day 15) ***		
FLR: <0.1%	20	27.8
FMR: 0.1– < 10%	37	51.4
FHR: ≥10%	12	16.7
N/A	3	4.2
**FC-MRD (Day 33) ***		
<0.1%	53	73.6
0.1– < 10%	7	9.7
≥10%	3	4.2
N/A	9	12.5

* The percentage of lymphoblasts were determined by conventional flow cytometry analysis.

**Table 2 cancers-13-05044-t002:** The detailed immunophenotype of lymphoblasts at pre-defined positions in a multidimensional dot-plots based on analysis used bivariate dot-plots by Tube 1 (A) and Tube 2 (B).

(A)							
		**Position 1**	**Position 2**	**Position 3**	**Position 4**	**Position 5**	**Position 6**
**CD66c**	negative	3.70%	93.30%	33.30%	100%	75%	66.70%
	weak positive	51.90%	6.70%	0%	0%	0%	33.30%
	positive	55.60%	0%	66.70%	0%	25%	0%
	bright positive	0%	0%	0%	0%	0%	0%
**CD34**	negative	11.10%	33.30%	0%	66.70%	0%	100%
	weak positive	7.40%	40%	0%	33.30%	0%	0%
	positive	74.10%	26.70%	100%	0%	100%	0%
	bright positive	7.40%	0%	0%	0%	0%	0%
**CD10**	negative	0%	0%	0%	0%	25%	0%
	weak positive	7.40%	0%	0%	0%	75%	0%
	positive	66.70%	0%	16.70%	0%	0%	100%
	bright positive	25.90%	100%	83.30%	100%	0%	0%
**CD38**	negative	25.90%	13.30%	16.70%	33.30%	0%	0%
	weak positive	25.90%	0%	16.70%	0%	0%	33.30%
	positive	48.20%	86.70%	66.70%	66.70%	100%	66.70%
	bright positive	0%	0%	0%	0%	0%	0%
**CD45**	negative	11.10%	26.70%	33.30%	66.70%	0%	0%
	weak positive	88.90%	0%	66.70%	33.30%	100%	100%
	positive	0%	73.30%	0%	0%	0%	0%
	bright positive	0%	0%	0%	0%	0%	0%
(B)							
		**Position 1**		**Position 2**	**Position 3**		**Position 4**
**CD58**	negative	6.30%		0%	0.0%		8%
	weak positive	37.5%		0.0%	14.3%		28%
	positive	56.3%		100.0%	85.7%		64%
	bright positive	0.0%		0.0%	0.0%		0%
**CD123**	negative	87.5%		67.5%	28.6%		48%
	weak positive	12.5%		33.3%	28.6%		16%
	positive	0.0%		0.0%	42.8%		36%
	bright positive	0.0%		0.0%	0.0%		0%
**CD33**	negative	62.5%		100.0%	71.4%		80%
	weak positive	37.5%		0.0%	28.6%		16%
	positive	0.0%		0.0%	0.0%		4%
	bright positive	0.0%		0.0%	0.0%		0%
**CD81**	negative	0.0%		0.0%	0.0%		0%
	weak positive	6.3%		0.0%	7.2%		0%
	positive	93.8%		100.0%	71.4%		36%
	bright positive	0.0%		0.0%	21.4%		64%

## Data Availability

Data represent confidential, patient-specific pieces of information. Data are available from the first and corresponding author (BK) upon reasonable request.

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
