# Peer review of "A Novel Method for the Evaluation of Bone Marrow Samples from Patients with Pediatric B-Cell Acute Lymphoblastic Leukemia—Multidimensional Flow Cytometry"

_cancers, 2021, doi:10.3390/cancers13205044_

Round 1
Reviewer 1 Report
In the manuscript "A novel method for the evaluation of bone marrow samples from patients with pediatric B-cell acute lymphoblastic leukemia - multidimensional flow cytometry" the authors investigate the key role of multicolor Flow Cytometry (FC) in acute lymphoblastic leukemia (ALL) diagnosis and prognosis. Particularly, they created a new analysis protocol using multidimensional dot-plots to characterize the pathological cell population at the time of diagnosis, to look for the association between immunophenotype and genotype, and to evaluate minimal residual disease (MRD).
The results showed that, in the determination of MRD, the new FC analysis can be considered as accurate as the conventional method since no remarkable differences between the multidimensional or conventional FC method at day 15 and day 33 were detected. Moreover, the multidimensional method is much faster and considerably reduces subjectivity compared to the analysis of two-dimensional dot-plots. Furthermore, the authors found associations between blasts position and their cytogenetic alterations.
Specific comments
Materials and methods: line 102-103
In Table 1 there are sixty-nine children who received appropriate bone marrow samples for the FC examination on day 15, not sixty-eight as reported in the text.
Materials and methods: line 104
It would be better to specify the gate where to acquire at least 30000 events
Materials and methods: line 115-116
The authors should specify the clones of monoclonal antibodies used to set up the multiparameters of tube 1 and tube 2. If is not possible, the authors could consider this information for the future.
Materials and methods: line 128-129
It would be more clear if the authors show the gate performed on normal and pathological B cells on the two-dimensional dot plots
Materials and methods: line 138-139
The authors should report detailed immunophenotype of both populations, normal and pathological
Materials and methods: line 146-147
It would be better if the authors explain clearly what criterion did they use to separate the radar dot-plot area into six parts in Tube 1 (Figure 2a) and four parts in Tube 2 (Figure 2b)
Materials and methods: line 167
The authors should replace dot-pots with dot-plots
Results: line 193-195
The authors should specify which marker, if changed in the panel, don’t affect the discrimination between normal and pathological B cells
The study is interesting and it could be useful to extend it by involving more centers and increasing the number of cases.
Author Response
We would like to thank Reviewer 1 for the useful suggestions. Based on those suggestions we have modified the manuscript. Below you will find please our responses to your comments in a point-by-point fashion.
Reviewer 2 Report
Kárai et al. redefine the analyzing protocols of conventional FACS analysis to understand and characterize the pathological cell populations by non-conventional multidimensional flow analysis. The study suggests that multidimensional dot plots flow analysis can be useful to analyze normal and pathological cells simultaneously, fast and accurate as compared to conventional Flow analysis. Experimental setup of multicolour labelled antibodies can be analyzed in a single tube and arrange as a single dot plot contrary to conventional analysis. In this study, they have analyzed the pediatric case of acute lymphoblastic leukaemia of 72 patients and explored the cytogenetic and immunophenotypic alterations. They also determined the population of lymphoblast and abnormal cells in minimal residual disease by this protocol.
Comments:
1- Kárai et al. conducted BM analysis of children with ALL and arranged the data on radar dot- plots to discriminate between normal and pathological cells, but the sample processing did not explain the selection of 4 samples (one normal and three pathological) for standardization.
2- CD19 marker is being used to determine B cells in BM samples. CD19 is a pre-B ALL marker with 100 % prevalence. The authors did not mention the initial screening of lymphoblast in BM and the significance of chosen SSC in ALL samples.
3- The authors showed that position and number of cells could distinguish the pathological vs normal population. Can authors show this by specific experimental data?
4- Day 15 and Day 33 variations should be more elaborated in the results sections than de novo. It should be explained why the number of cells dynamically changed in these time points in normal and pathological cells.
5- Data representation and results of Fig 1 and 2 is unclear. The authors should mention the outcomes in a better way to justify the claims mentioned.
Author Response

(The authors gave the same response as above.)

Reviewer 3 Report
A novel method for the evaluation of bone marrow samples from patients with pediatric B-cell acute lymphoblastic leukemia - multidimensional flow cytometry
1- I know you have defined the word FC in the simple abstract but please redefine this in the actual abstract since using undefined acronyms in the abstract is not advised.
2- Possibly substitute the word profile for appearance in line 28
3- The abstract is hard to follow since not enough detail is given about this multidimensional dot-plot method but rather a lot of information about how much better it is compared to previous methods which seems something that belongs in the discussion. The abstract needs rewording.
4- What is the significance of this sentence: “Furthermore, we found associations between the position and the number of clusters of blast cells and cytogenetic properties (p=0.002 and p<0.0001 by the position and p=0.02 35 by the number of subclones).”
5- Quick is colloquial, you might want to use rapid instead.
6- “The absence of CD10 marker on abnormal lymphoblasts is associated with KMT2A gene rearrangement, and CD66c and CD25 positivity is associated with the presence of Philadelphia chromosome in ALL cases”. These are useful links but how about some genetic alterations that may not be associated with an FC-detectable marker?
7- it is as sensitive, I think it should be a, in line 66.
8- “Briefly, ‘B-other’ group includes patients without recurrent genetic abnormalities, such as t(9;22)/BCR-ABL1, KMT2A (MLL) rearrangements, t(12;21)/TEL-AML1, t(1;19)/TCF3-PBX1 or high hyperdiploidy”. So, is the new method useful for MRD for both groups of patients B-other and otherwise?
9- “whether each antigen is expressed or not, and even if its presence is dim or bright”. If you are referring to the bivariate dot-plots with respect to the binary option of the presence or absence of the antigen then how does this bivariation account for different levels of colour (dim or bright).
10- What does an interpreter refer to, is this the software or specialist staff? At any rate, AI diagnostics must always be validated by specialist medical staff.
11- “In case of MRD detection, it is also a challenge to identify the abnormal cells in the regenerating bone marrow (BM), where all the stages of normal maturation are present during therapy; moreover, an immunophenotype shift of abnormal cells can also occur”. This may then require single-cell analysis.
12- In line 88, since your topic is a new method, it is important that you give more emphasis to the findings of previous papers mentioned in [26-28]. I would add more detail about their findings to better support your study and also make it clear to the readers what your study has to offer over previous reports.
13- Please explain what you mean by appropriate samples in line 103, it is currently vague. Is this related to the criteria you have defined in 103-106?
14- Since these were BM samples, were these samples purified? Since as you know much debris would be present in a BM extract. If so, please provide your methods.
15- Is the information detailed in table 1, especially the cytogenetic information and FC-MRD established using other methods and used as a reference for this study? If so, kindly make this clear. How has this previous information assisted your current study? How well has your method performed?
16- Line 124, is this the new method that this paper is referring to? It’s not clear.
17- “First, data of a normal BM sample and three BM samples of patients with ALL were merged.” Can you explain why this was done? I assume there were individual ALL and normal tubes as well, so that, through comparison, one can deduce where the normal and pathological samples were located?
18- “and those parameters and locations were selected in which the normal and pathological cell populations differed from each other the most”. Line 130, a graphical representation would help appreciate this better and it will help reproducibility.
“Finally, lymphoblasts gated on radar dot-plots were displayed on bivariate dot-plots to examine the expression of certain markers individually.” This again is vague language, why use certain, when you can name the marker.
“The detailed immunophenotype was necessary in order to test that the population with a position different from normal cells was really the pathological population.” Yes, but what was this phenotype-position, please provide detail.
19- I am completely confused about what I am looking at in figure 1. The text pertaining to figure 1 does not define tube 1 and tube 2 or de novo samples but explains the method of gating (lines 133-137). Please assign immunophenotypes to each of the positions shown in figure 1.
20- What were the contents of tubes 1 and 2 in figure 2? Please assign immunophenotypes to each of the positions shown in figure 2.
21- “Then we compared the position of lymphoblasts with cytogenetic alterations. Patients with KMT2A gene rearrangement, iAMP21, t(1;9), and t(1;19) were excluded from the study due to the low number of cases”. Why was the number significant here?
22- 160-164- are the authors showing any data for this?
23- The explanation for figure 3 is insufficient, since this is a methods paper, it needs much more detail.
24- The p in p-value must be italicised throughout.
25- Tube 1 and 2 in figure 4 need to be defined.
26- In figure 5, I realise that the 6 and 4 positions relate to figure 2, but what is the immuno-identity of each of these positions? Also, since the identity of the contents of the tubes is not clear, I am not sure what I am looking at or reading. Exact phenotypes for each position must be defined.
27- In general this method paper has not been described in a comprehensible way and requires much more clarification and detail. How was the proposed method compared to previous methods is unclear?
28- “Then the detected immunophenotype of pathological cells is compared to the learned patterns”. This is the first time that previous methods have been discussed.
29- “In case of multidimensional dot plots, it is not required to know all expression changes of antigens during maturation, it is enough to learn the position of normal cell populations.” Again, this is very subjective since you have neither defined the phenotype of cells in each position nor have you offered a clear and reproducible way to gate. None of the samples referred to were sufficiently described. Pictures of the process would help.
30- Figure 3, what are these cells and how are they heterogeneous.
31- For figure 4, what comparisons have been made to deduce these results?
32- What does the finding of figure 6 signify?
Author Response

(The authors gave the same response as above.)

Round 2
Reviewer 3 Report
The authors have addressed my comments.